# False Information Detection via Multimodal Feature Fusion and Multi-Classifier Hybrid Prediction

Yi Liang, Turdi Tohti * 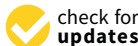 and Askar Hamdulla 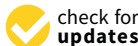

School of Cyber Science and Engineering, College of Information Science and Engineering, Xinjiang University, Urumqi 830017, China; liangyi@stu.xju.edu.cn (Y.L.); askar@xju.edu.cn (A.H.)
* Correspondence: turdy@xju.edu.cn; Tel.: +86-139-9999-4696

**Abstract:** In the existing false information detection methods, the quality of the extracted single-modality features is low, the information between different modalities cannot be fully fused, and the original information will be lost when the information of different modalities is fused. This paper proposes a false information detection via multimodal feature fusion and multi-classifier hybrid prediction. In this method, first, bidirectional encoder representations for transformers are used to extract the text features, and S win-transformer is used to extract the picture features, and then, the trained deep autoencoder is used as an early fusion method of multimodal features to fuse text features and visual features, and the low-dimensional features are taken as the joint features of the multimodalities. The original features of each modality are concatenated into the joint features to reduce the loss of original information. Finally, the text features, image features and joint features are processed by three classifiers to obtain three probability distributions, and the three probability distributions are added proportionally to obtain the final prediction result. Compared with the attention-based multimodal factorized bilinear pooling, the model achieves 4.3% and 1.2% improvement in accuracy on Weibo dataset and Twitter dataset. The experimental results show that the proposed model can effectively integrate multimodal information and improve the accuracy of false information detection.

**Keywords:** false information detection; hybrid fusion method; computer vision; deep autoencoder

## 1. Introduction

With the development of social networks, the speed of information dissemination is increasing rapidly, which not only facilitates people to socialize but also facilitates the spread of false information [1,2]. False information seriously damages the credibility of the media, infringes the public's right to know, hinders the public's right to participate and supervise, and in severe cases can disrupt social order, cause people's property damage, cause public panic, disrupt social order and have serious adverse effects on society. For example, in 2011, the rumor that iodized salt would prevent nuclear radiation caused by the Fukushima nuclear accident in Japan caused everyone to frantically buy salt, which not only caused a waste of resources but also disturbed social stability. Therefore, how to detect false information at an early stage has become a recent research hotspot. Early information is in the form of plain text, so the early methods mainly detect false information by extracting text features from text content or extracting corresponding single-modality features from other single-modality data [3,4]. With the development of social media, the form of information has changed from the form of plain text to the form of multimedia [5–7]. Most of the existing information is in the form of multimedia. At the same time, research found that the fusion of features of different modalities can effectively improve the performance of the model. Therefore, the recent research in the field of false information detection is mainly based on multimodal methods. However, the existing multimodal methods have shortcomings. First, most of the existing multimodal methods use early

fusion methods or late fusion methods. However, there are many limitations in using only one method to fuse multimodal information, and it is difficult to fully fuse the information of different modalities. Second, for text feature extraction, most of them rely on the concatenated output of bi-gated recurrent unit (Bi-GRU) at each time step. However, the feature extraction process lacks the participation of corresponding factual knowledge. Such methods have limited ability to understand named entities in the text, and thus, it is difficult to fully capture the clues at the semantic level of false information [8]. The extraction of image features mostly relies on VGG19 [9] to obtain features. This method requires a large number of parameters and computing resources during training. The quality of features extracted by these two models is lower than that of some existing models. Finally, the problem of original information loss is not considered when performing information fusion between modalities.

In this paper, the above-mentioned issues are investigated separately:

(1) This paper studies the problem of how to mitigate the influence of irrelevant features and how to enhance fusion between multimodalities. In this paper, the deep autoencoder is used to fuse the information of different modalities early, and the noise removal function of the deep autoencoder is used to reduce the impact of noise on the model performance. The multi-classifier hybrid prediction method is used to fuse the multimodal information in the later stage to enhance the fusion effect between different modalities.

(2) For the problem of low quality of single-modality features, this paper uses bidirectional encoder representations for transformers (BERT) [10] to replace Bi-GRU in the text feature extraction. BERT has proven its effectiveness in many fields. Shifted window transformer (SWTR) [11] is used to replace VGG19 in image feature extraction. SWTR has achieved good performance in multiple tasks, such as object detection, instance segmentation and semantic segmentation since its release. In this paper, SWTR is applied to multimodal false information detection tasks as an image feature extraction module. The experimental results show that SWTR also has a good performance in the field of false information detection.

(3) For the problem that the deep autoencoder will lose some information while removing noise, after the joint feature is obtained by the deep autoencoder, the text feature and image feature are spliced with the feature to reduce the loss of original information.

(4) Comparative experiments and ablation experiments are performed on the Chinese and English datasets, which prove the effectiveness of the model and its various modules in false information detection.

## 2. Related Work

False information is defined as unsubstantiated or intentionally fabricated stories or statements [12]. Multimodal data refers to the description of different angles of the same object; the description of each angle is a modality [13]. This paper divides the current research into two categories according to the number of modalities used: methods based on single modality and methods based on multimodality [14].

### 2.1. Single-Modality-Based Approach

Content features refer to features that can be extracted directly from published information. Content features mainly include text features and visual features [15].

In the early research, people's processing of text content is mainly through manual extraction of text features and the use of machine-learning classification algorithms to detect false information. The method based on text features detects posts by extracting features such as text length, word frequency, part-of-speech and word vectors, morphological and syntactic features from the text information of a post. Verónica [16] et al. built a fake news detector based on linguistic features. The authors extract N grams, punctuation, psycholinguistic features, features indicating text intelligibility and grammar into five sets of linguistic features from the text and finally use the SVM classifier and five cross-

validations for detection. Kwon [17] used structural information and linguistic features to capture the multimodal phenomenon of rumor propagation using three classification models: support vector machine, random forest classifier and decision tree. In the early days of the multimedia age, fake news began to use multimedia content with pictures to attract readers to spread false information. Jin [18] et al. demonstrated the importance of images in fake news detection. Due to lack of expertise, it is difficult for traditional machine-learning-based false information detection models to obtain handcrafted features.

With the widespread application of deep learning, Ma [19] et al. used a recurrent neural network to obtain the hidden features of text, which were used to detect rumors. Qi [20] et al. used an attention mechanism to dynamically fuse the features of the image frequency domain and pixel domain for false information detection. Although deep learning can automatically extract features, it is often affected by noise, which degrades its model performance. Inspired by generative adversarial networks (GAN), Ma [21] et al. proposed a model based on GAN network, which can remove noise and irrelevant features through adversarial training to obtain more discriminative features.

Although the detection of false information using text features or visual features alone has been proved to be effective, with the advent of the multimedia era, the existing form of information has changed from the form of plain text in the past to the form that contains text, pictures, videos and other multimodal data at the same time, so if false information is detected only from the perspective of text or pictures, not only are the information utilization ratio and detection accuracy low, but it is also difficult to adapt to the multimedia era that had arrived.

### 2.2. Multimodal-Based Approach

Most of the existing multimodal-based methods use two kinds of modal information, text and picture, to detect false information. Singhal [22] et al. used BERT to extract text features and VGG19 to extract image features, which were concatenated as joint features for false information detection. Kumari [23] et al. proposed a multimodal fusion model based on multimodal decomposition bilinear pooling. This method can solve the problem that the simple concatenating of different features cannot determine the feature boundary, and the correlation between image feature and text feature cannot be found.

Although the above studies have proposed different ideas in the fusion module, the input is only two modalities of text and image, ignoring the role of the other modalities. Giachanou [24] et al. extract sentiment from text while extracting features from text, extract image labels when extracting image features and perform similarity calculation between image labels and text features. Through the processing of the above method, more information can be obtained from the text and images, and the utilization ratio of the information can be increased. However, it only performs feature fusion through simple concatenating, which fails to fully consider the correlation between different modalities. YUAN [8] et al. proposed semantically enhanced multimodal false information detection, which obtains image features, graph labels and text in the graph through a convolutional network and updates the text information by adding the text extracted from the picture to the text information. The graph labels, original text information and updated text information are used to extract features through enhanced representation from knowledge integration (ERNIE) [25]. The graph labels and graph feature vectors are updated through an attention mechanism using features extracted from the original text. Finally, the updated text features, the updated graph label features and the updated image features are concatenated as the final joint feature for detection.

Meng et al. [14] proposed a multimodal fusion model based on attention mechanism. The model first uses Bi-GRU to extract semantic features and uses convolutional recurrent neural network to extract the features of different levels of the picture and then uses the inter-modality attention and the intra-modality attention to fuse the multimodal information to obtain joint features. Finally, the text features, image features and joint features are fused through the attention mechanism to enhance the role of the original information.

The above research works use simple splicing or attention methods to fuse text features and image features, and they only perform early fusion of the two features, ignoring the effect of later fusion [26]. At the same time, in the early fusion, the influence of noise on the fusion results is not considered [27]. In this paper, the above problems are studied, and a deep-autoencoder-based false information detection method is proposed.

## 3. False Information Detection Method

Problem Definition: Suppose $P = \{p_1, p_2, \ldots, p_m\}$ is a dataset of multimodal posts in social networks where $p_i$ is the ith post. $T = \{t_1, t_2, \ldots, t_m\}$ is the text set, $t_i$ is the text content in the ith post. $V = \{v_1, v_2, \ldots, v_m\}$ is an image set, $v_i$ is the image included in the ith post. $L = \{l_1, l_2, \ldots, l_m\}$ is the tag set, $l_i$ is the tag of the *i*th post. $p_i = \{t_i, v_i, l_i\}$, The task of false information detection can be described as learning a function $f(T, V) = Y$, $Y = \{y_1, y_2, \ldots, y_m\}$, $y_i$ is the predicted label value of the *i*th post, $y \in \{0, 1\}$, 0 represents true information, and 1 represents false information.

The model in this paper is mainly composed of four parts: text feature extractor, image feature extractor, early fusion module and late fusion module. Figure 1 shows our proposed false information detection method based on multimodal feature fusion and multi-classifier hybrid prediction.

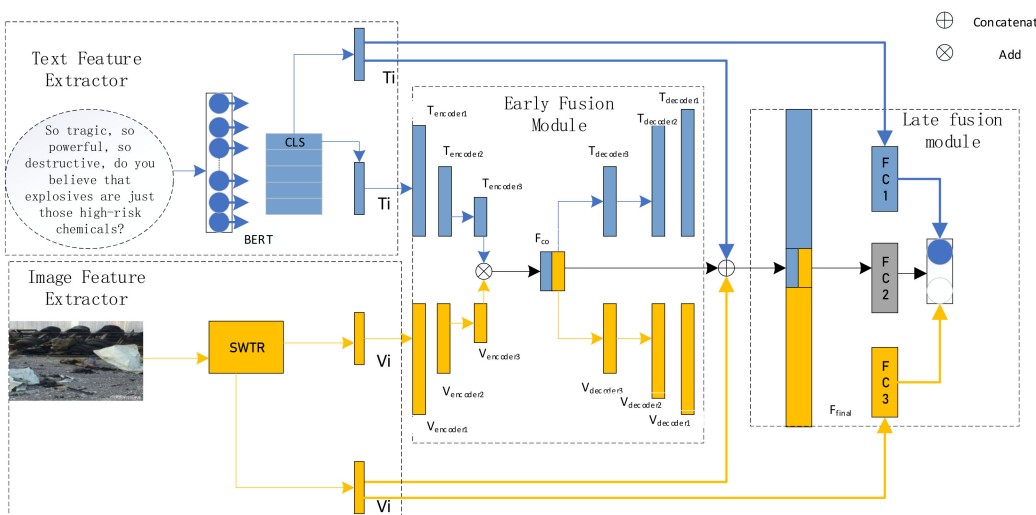

**Figure 1.** Our false information detection model.

The model first uses the text feature extractor and the image feature extractor to extract the text and image features, and then input the two obtained features into the early fusion module to obtain joint features, and the text features, image features and joint features are concatenated to strengthen the joint features. Finally, the text features, the image features and the enhanced joint features are used to obtain the prediction results through the later fusion module. The early fusion module realizes early fusion between different modal features through the deep autoencoder, and the late fusion module uses the method of multi-classifier mixed prediction to fuse the prediction probabilities of different modalities, The pseudocode is shown in Table 1.

### 3.1. Text Feature Extraction

The text is the main part of the post. It contains the main information that the publisher wants to express, and it can also show the publisher's emotions and other information. We adopt the BERT pre-training model with strong modeling ability to deal with this important part.

**Table 1.** Whole pseudocode.

| | |
|---|---|
| Input : $t_i, v_i$ | #$t_i, v_i$ represents text and image content in a message |
| Output : $y_i$ | #$y_i$ is the prediction result of the model for this information |
| $T_i = BERT(t_i) \in R^{1*768}$ | #$T_i$ is text feature |
| $V_i = Swin - Transformer(v_i) \in R^{1*768}$ | #$V_i$ is image feature |
| $F_{co} = Deep\_autoencoder(T_i, V_i) \in R^{1*64}$ | #$F_{co}$ is fused feature |
| $F_{final} = concate(T_i, V_i, F_{co}) \in R^{1*1600}$ | #concatenation $T_i, V_i$ and $F_{co}$ |
| $P_{text} = FC1(T_I) \in R^{1*2}$ | #$P_{text}$ is the predicted probability through the text feature $T_i$ |
| $P_{co} = FC2\left(F_{final}\right) \in R^{1*2}$ | #$P_{co}$ is the predicted probability through the fused feature $F_{final}$ |
| $P_{image} = FC3(V_i) \in R^{1*2}$ | #$P_{image}$ is the predicted probability through the image feature $V_i$ |
| $P_{final} = w_1 p_{text} + w_2 p_{co} + w_3 p_{image} \in R^{1*2}$ | #$P_{final}$ is the final predicted probability |
| $y_i = argmax\left(P_{final}\right)$ | #$y_i$ is the predicted result |

BERT is a pre-training model based on Transformer [28]. Its special unsupervised task enables it to learn contextual information, and it only uses self-attention mechanism instead of RNNs so that it can be parallelized to speed up the training process, and because it has enough parameters, the model can learn more knowledge through learning on large-scale pre-training corpus. It learns some syntactic knowledge and common sense. In some recent similar tasks, usually, BERT-based models will perform better than other networks built on RNNs and CNNs. These experiments show that higher-quality text features can be extracted using BERT. Therefore, this paper uses BERT to extract text features. BERT is used to input text in the form of '[CLS]' + sentence + '[SEP]', and the output is an embedded representation of a set of words. The text features are calculated as follows:

$$T_i = Bert(t_i) \tag{1}$$

where $T_i$ is a collection of text, $T_i \in R^{d_i}$ is the feature vector of the text, $d_i$ is the dimension of the feature vector.

Since BERT uses the self-attention mechanism internally, if a word with a specific meaning is used to represent the entire sentence, the word vector will be affected by the word itself, and it is difficult to objectively represent the characteristics of the entire sentence. Since the [CLS] vector is a marker bit and does not have semantic information, it will not be affected by itself, so it can be better used as a feature of the entire text. Therefore, this paper selects the [CLS] vector of each sentence to represent the features of the entire text.

*3.2. Image Feature Extraction*

Picture can supplement the text information and increase the credibility of the content. Processing image parts can better understand the semantics of multimodal posts and thus better detect false information. We use SWTR to process the image part.

SWTR is also an image feature extraction model based on Transformer. This model expands the applicability of Transformer, transfers the high performance of Transformer to the visual neighborhood, solves the shortage of CNN for global information feature extraction, and at the same time, with its unique window mechanism, it greatly reduces the computational complexity of self-attention and solves the problem of fixed token scale, which has become the general backbone of computer vision. Since it was proposed, it has achieved relatively good results in tasks, such as image classification and segmentation tasks. Therefore, this paper uses SWTR as the image feature extractor. The input of *SWTR* is a picture, and the output is the feature vector of the picture. The specific calculation process is as follows:

$$V_i = SWTR(I_i) \tag{2}$$

where $I_i$ is a picture contained in the ith post, $V_i \in R^{d_i}$ is the feature vector of the picture, and $d^i$ is the dimension of the feature vector.

### 3.3. Early Fusion

We obtained text features $T_i$ and image features $V_i$. In order to obtain the interactive relationship between the text and the image, this paper uses a deep autoencoder to interact with text and image features to obtain multimodal joint features. The deep autoencoder will convert the high-dimensional feature vector into a low-dimensional feature vector through a function. The low-dimensional feature vector will contain most of the main features of the data, and some tiny features may be lost, so as to eliminate irrelevant features in the features. This module firstly inputs the features of text and images into the encoder for encoding. After the features of text and images enter the encoder, they will be compressed into low-dimensional feature vectors through two linear layers, and then add the encoded two feature vectors by dimension. Finally, the added feature vector is input into the two decoders to obtain two outputs. The specific calculation process is as follows:

$$T_{encoder1} = Linear(T_i) \tag{3}$$

$$T_{encoder2} = Linear(T_{encoder1}) \tag{4}$$

$$T_{encoder3} = Linear(T_{encoder2}) \tag{5}$$

$$V_{encoder1} = Linear(V_i) \tag{6}$$

$$V_{encoder2} = Linear(V_{encoder1}) \tag{7}$$

$$V_{encoder3} = Linear(V_{encoder2}) \tag{8}$$

$$F_{co} = add(T_{encoder}, V_{encoder}) \tag{9}$$

$$T_{decoder3} = Linear(F_{co}) \tag{10}$$

$$T_{decoder2} = Linear(T_{decoder3}) \tag{11}$$

$$T_{decoder1} = Linear(T_{decoder2}) \tag{12}$$

$$V_{decoder3} = Linear(F_{co}) \tag{13}$$

$$V_{decoder2} = Linear(V_{decoder3}) \tag{14}$$

$$V_{decoder1} = Linear(V_{decoder2}) \tag{15}$$

$Linear()$ is a linear transformation function, which transforms the dimension of the feature vector, and $add()$ is an addition operation, which adds the two input feature vectors according to the corresponding dimensions. $T_{encoder} \in R^{e_i}$ is the feature of the text feature $T_i$ processed by the encoder, and $V_{encoder} \in R^{e_i}$ is the feature of the image feature $V_i$ processed by the encoder. $T_{encoder}, V_{encoder}, F_{co} \in R^{e_i}$, $e_i$ is the dimension of the feature vector processed by the encoder. $T_{decoder}, V_{decoder} \in R^{d_i}$ is the feature vector decoded by the $T_{encoder}$, and $V_{encoder}$, $d_i$ is the dimension of the feature vector.

The $T_{encoder}$ and $V_{encoder}$ are only used when training this module. This module obtains the prediction error by comparing the output of the decoder with the input of the encoder, and then backpropagation is performed to train the deep autoencoder and gradually improve the accuracy of autoencoding, and finally, the trained model is used as the early fusion module of the model in this paper. In this paper, the $F_{co}$ after the addition of $T_{encoder}$ and $V_{encoder}$ is taken as the joint features of multimodality.

### 3.4. Late Fusion

Because the deep autoencoder changes the feature vector from high dimensional to low dimensional, it not only removes some irrelevant features but also loses some original information of text and images. Therefore, this paper does not use the joint features $F_{co}$ extracted from the early fusion module to train classifier 2 but uses the joint features $F_{final}$ enhanced by the original information to train it. The enhanced joint features $F_{final}$ is

obtained by concatenating the text features $T_i$ and image feature $V_i$ with the multimodal joint feature $F_{co}$.

$$F_{final} = T_i \oplus F_{CO} \oplus V_i \tag{16}$$

$\oplus$ is the concatenating operation, $F_{final} = [T_i, F_{co}, V_i]$.

The three classifiers, $FC_1$, $FC_2$ and $FC_3$, are all fully connected layers with activation function $softmax$. $FC_1$ classifies the true and false information through text information, $FC_2$ classifies the true and false information through the fusion information of text and images, and $FC_3$ classifies the true and false information through image information. The text feature $T_i$, the enhanced joint features $F_{final}$ and the image feature $V_i$ are projected to the binary target space to obtain the probability distributions $P_{text}$, $P_{co}$ and $P_{image}$.

$$P_{text} = softmax(W_t T_i + b_t) \tag{17}$$

$$P_{co} = softmax(W_c F_{final} + b_c) \tag{18}$$

$$P_{image} = softmax(W_v V_i + b_v) \tag{19}$$

where $W_t$, $W_c$ and $W_v$ represent weight parameters, and $b_t$, $b_c$ and $b_v$ represent bias terms.

Then, the obtained three probability distributions are fused in proportion to obtain the final predicted probability.

$$P_{final} = w_1 p_{co} + w_2 p_{text} + w_e p_{image} \tag{20}$$

$$w_1 + w_2 + w_3 = 1 \tag{21}$$

where $w_1$, $w_2$ and $w_3$, respectively, represent the degree of influence of different parts of the model on the final detection results.

*3.5. False Information Detection*

After normalizing the probability distribution of the later fusion by the function, take out the one with the highest probability as the final result.

$$P = softmax(P_{final}) \tag{22}$$

$$y_i = \text{argmax}(P) \tag{23}$$

$y_i$ is the predicted tag value of our model for *i*th post.

The loss function is defined as the cross-entropy loss function between the predicted probability distribution and the true label:

$$L = -\sum_{i=1}^{m} [l_i \log p_i + (1 - l_i) \log(1 - p_i)] \tag{24}$$

where $m$ is the number of posts, which $l_i \in \{0, 1\}$ is the true label value, 1 represents false information, 0 represents true information, and $p_i$ represents the probability of being predicted to be false information.

**4. Experimental Results and Analysis**

*4.1. Experiments*

Machine configuration and environment for this experiment: CPU: Intel Xeon E5-2630L v3, 62 G memory, 8cores, GPU: NVIDIA GeForce RTX 3090, PyTorch(1.7.1), Python(3.8), Cuda(10.2). In our experiments, we use two publicly available datasets, namely Twitter and Weibo. The selected Twitter dataset is a public dataset: https://github.com/MKLab-ITI/image-verification-corpus (accessed on 18 October 2021). The selected Weibo dataset is a public dataset: https://drive.google.com/file/d/14VQ7EWPiFeGzxp3XC2DeEHiBEisDINn/view?usp=sharing (accessed on 18 October 2021). The Twitter dataset contains text, related images and contextual information. The information in the training and test sets is collected from different events, so there is no temporal overlap between the training and test sets. This training set contains 14,483 tweets divided into three categories: false, true and humorous. A total of 6841 of them are fake, 5009 are true, and 2633 are

humorous. The test set contains 3781 tweets. Since the test set does not contain humor class instances, we ignore these instances in our experiments. The Twitter dataset contains 360 training set images and 50 test set images. We use 20% of the training set as the validation set [18]. The Weibo dataset contains a total of 9563 news items. Of these, 4784 were false and 4779 were true. The test dataset includes 7567 messages. We use 20% as the validation dataset, and the test dataset includes 1996 messages, Table 2 shows the complete data distribution of the two data sets. All the hyper-parameters used for training the proposed model are listed in Table 3.

**Table 2.** The dataset used in the experiment.

| Data | Train | | Test | | Image |
|---|---|---|---|---|---|
| | **Fake** | **Real** | **Fake** | **Real** | |
| Twitter | 6841 | 5009 | 2564 | 1217 | 410 |
| Delete_b | 3784 | 3783 | 1000 | 996 | 13,274 |

**Table 3.** Parameter settings.

| Parameters | Twitter | Weibo |
|---|---|---|
| Text length | 32 | 95 |
| Image size | (2,242,243) | (2,242,243) |
| Batch size | 128 | 32 |
| Optimizer | BertAdam (lr = $5 \times 10^{-5}$) | BertAdam (lr = $5 \times 10^{-5}$) |
| Epochs | 40 | 40 |
| Dropout | 0.6 | 0.6 |
| $w_{text}$ | 0.53 | 0.48 |
| $w_{co}$ | 0.03 | 0.07 |
| $w_{image}$ | 0.44 | 0.45 |

*4.2. Evaluation Index*

For binary classification problems, we generally use accuracy, precision, recall and F1 value to evaluate the performance of the model. The accuracy rate is the proportion of all predictions that are correct to the total. The accuracy rate is the proportion of all positive predictions that are correctly predicted. The recall rate is the proportion of correct predictions that are positive to all actual positives. The $F_1$ value is an overall evaluation of precision and recall. The calculation formulas of *Accuracy*, *Precision*, *Recall* and $F_1$ value are described as follows:

$$Accuracy = \frac{TP + TN}{TP + TN + FP + FN} \tag{25}$$

$$Precision = \frac{TP}{TP + FP} \tag{26}$$

$$Recall = \frac{TP}{TP + FN} \tag{27}$$

$$F_1 = \frac{2TP}{2TP + FP + FN} \tag{28}$$

The specific meanings of *TP*, *TN*, *FP* and *FN* are shown in Table 4. *TP* is a positive sample predicted by the model as a positive class. *TN* is a negative sample predicted by the model to be a negative class. *FP* is a negative sample predicted by the model as a positive class. *FN* is a positive sample predicted by the model as a negative class.

*4.3. Baseline Model*

We implement some baseline models based on single-modality and multimodal methods to verify the effectiveness of our proposed model.

**Table 4.** Meaning of *TP*, *FP*, *TN*, *FN*.

| Reality | Predict Result | |
| --- | --- | --- |
| | **Positive Example** | **Negative Example** |
| Positive example | *TP* | *FN* |
| Negative example | *FP* | *TN* |

4.3.1. Single-Modality Model

(1) Text: First input the text content into the trained BERT pre-training model to obtain the word embedding representation of the word. Then, extract the [CLS] vector output by BERT, and then, input it into a fully connected layer whose activation function is *softmax* for classification.

(2) Visual: Input the image into the trained SWTR model to obtain the feature vector of the image, and then, input it into a fully connected layer whose activation function is *softmax* for classification.

4.3.2. Multimodal Model

(1) att-RNN [29]: att-RNN is to use RNN with attention mechanism to fuse text, image and social context features to achieve the task of rumor detection.

(2) EANN [30]: Even adversarial neural network (EANN) has three main components: multimodal feature extractor, false information detector, event discriminator. In the model, the feature extractor will extract the textual and visual features of the event with the help of the event discriminator and then concatenate the obtained textual and visual features and finally use the false information detector to detect the authenticity of news posts.

(3) MVAE [31]: Multimodal variational autoencoder (MVAE) trains three sub-networks to detect fake news. Here, a variational autoencoder is trained for better representation of textual and visual features. The shared latent features are further used for classification.

(4) AMFB [23]: Attention-based multimodal factorized bilinear (AMFB). The network consists of three parts, a feature extraction module, a feature fusion module and a false information detection module. Attention-based multimodal factorization bilinear pooling to fuse text and image features.

(5) OUR1: The false information detection model proposed in this paper consists of a feature extractor, an image feature extractor, an early fusion module and a late fusion module. The later fusion module of the model adopts the method of adding the probability distributions obtained by the three classifiers in proportion to obtain the final probability distribution and determine the result by the added probability distribution.

(6) OUR2: This model adopts the same method as OUR1 in text feature extractor, image feature extractor and early fusion module, but it adopts a voting method in the later fusion module to obtain the result. Each classifier can cast one vote, and the one with the most votes is the final result.

(7) OUR3: This model is the same as OUR2, only the late fusion module was changed. This model takes into account that the proportion of real information in real life is much larger than false information, so it adopts the 'one-vote veto' method in the later fusion module to obtain the results. The method first checks the classification results output by the three classifiers, and if one of them is true, the information is considered to be true.

*4.4. Analysis of Results*

The experimental results of the baseline models and our model on the two datasets are shown in Table 5. We report the accuracy, precision, recall and F1-score of our model. The experimental results show that the accuracy of our model is better than the baseline models on the Weibo dataset and the Twitter dataset.

**Table 5.** Experimental results on two datasets.

| Dataset | Model | Accuracy | Fake News | | | Real News | | |
|---------|-------|----------|-----------|--------|---------|-----------|--------|---------|
| | | | Precision | Recall | F-Score | Precision | Recall | F-Score |
| Twitter | Text | 0.820 | 0.856 | 0.858 | 0.857 | 0.758 | 0.756 | 0.757 |
| | Visual | 0.839 | 0.850 | 0.903 | 0.876 | 0.816 | 0.729 | 0.770 |
| | att-RNN | 0.664 | 0.749 | 0.615 | 0.676 | 0.589 | 0.728 | 0.651 |
| | EANN | 0.741 | 0.690 | 0.550 | 0.610 | 0.760 | 0.850 | 0.810 |
| | MVAE | 0.745 | 0.801 | 0.719 | 0.758 | 0.689 | 0.777 | 0.730 |
| | AMFB | 0.883 | 0.890 | 0.950 | 0.920 | 0.870 | 0.760 | 0.810 |
| | OUR1 | 0.895 | 0.938 | 0.892 | 0.914 | 0.831 | 0.900 | 0.864 |
| | OUR2 | 0.841 | 0.908 | 0.832 | 0.868 | 0.751 | 0.856 | 0.800 |
| | OUR3 | 0.756 | 0.728 | 0.977 | 0.834 | 0.907 | 0.382 | 0.538 |
| Weibo | Text | 0.851 | 0.829 | 0.899 | 0.863 | 0.880 | 0.800 | 0.838 |
| | Visual | 0.705 | 0.751 | 0.647 | 0.695 | 0.668 | 0.768 | 0.715 |
| | att-RNN | 0.772 | 0.797 | 0.713 | 0.692 | 0.684 | 0.840 | 0.754 |
| | EANN | 0.791 | 0.840 | 0.720 | 0.780 | 0.760 | 0.860 | 0.800 |
| | MVAE | 0.824 | 0.854 | 0.769 | 0.809 | 0.802 | 0.875 | 0.837 |
| | AMFB | 0.832 | 0.820 | 0.860 | 0.840 | 0.850 | 0.810 | 0.830 |
| | OUR1 | 0.875 | 0.850 | 0.921 | 0.884 | 0.906 | 0.825 | 0.863 |
| | OUR2 | 0.773 | 0.739 | 0.872 | 0.800 | 0.828 | 0.667 | 0.739 |
| | OUR3 | 0.597 | 0.564 | 0.994 | 0.719 | 0.965 | 0.167 | 0.284 |

By comparing the experimental results of the models using different late fusion methods, it can be found that the model using the proportional addition method for late fusion has higher accuracy, which shows that the effect of proportional addition is better than the two fusion strategies of voting and 'one-vote veto', so this paper uses OUR1 as the final false information detection model.

From the perspective of single modality and multimodality, the accuracy of the multi-modal model is higher than that of the single-modality model, indicating that multimodal models can obtain more informative features and improve the accuracy of model detection through complementary learning between modalities.

The accuracy of our model on the Twitter dataset is 1.2% higher than that of AMFB, 15% higher than that of MVAE and 15.4% higher than that of EANN. On the Weibo dataset, the accuracy of our model is 4.3% higher than that of AMFB, 5.1% higher than that of MVAE and 8.4% higher than that of EANN.

*4.5. Ablation Experiment*

In this paper, the effectiveness of each module is verified by comparing the original model with the model after deleting different modules. The results are shown in Tables 6 and 7:

(1) *OUR*: contains all modules.
(2) *Delete_t*: The late fusion module is removed, and the text features in the enhanced joint features are removed.
(3) *Delete_v*: The late fusion module is removed, and the image features in the enhanced joint features are removed.
(4) *Delete_c*: The late fusion module is removed, and the original joint features in the enhanced joint features are removed.
(5) *Delete_b*: Deleting the late fusion module.

We use 5-fold cross validation experiment to demonstrate the validity of our results. Due to the imbalanced distribution of the Twitter dataset, cross-validation cannot be performed well, so we only conduct experiments on the Weibo dataset; the results are shown in Table 6. The accuracy of each model in Table 6 is the average value after 5-fold cross-validation, and the standard deviation of each method is calculated. As can be seen from Tables 6 and 7, the performance of the model decreases when any module is removed. Compared with the original model, the *Delete_b* model has a 0.3% reduction in

the accuracy of the Weibo dataset and Twitter dataset, which indicates that the contributions of different modal features to the final result are not the same, and the multimodal features are only passed through the early stage, which is insufficient, and the information between the modalities can be fully fused by using the early fusion module and the late fusion module at the same time. Compared with the original model, the *Delete_c* model has lower accuracy on the Weibo dataset and Twitter dataset, indicating that the use of deep autoencoder for early fusion is effective. The accuracy rates of *Delete_t* and *Delete_v* also decrease on both datasets, indicating that in the process of using the deep autoencoder for early feature fusion, the unique denoising function of the deep autoencoder can not only remove irrelevant features but also lose some useful features. Both text information and image information are partially lost. This part of the loss can be reduced by splicing the original features. The loss of this part can be reduced by concatenating the original features. Comparing the performances of *Delete_t* and *Delete_v* on the two datasets, it can be found that deleting text features has a greater impact on model accuracy than deleting image features, indicating that text features will have more losses than image features during the fusion process.

**Table 6.** Ablation experiments on Weibo dataset.

| Model | Accuracy | Fake News | | | Real News | | | STDEV |
|---|---|---|---|---|---|---|---|---|
| | | Precision | Recall | F-Score | Precision | Recall | F-Score | |
| *OUR* | 0.875 | 0.850 | 0.921 | 0.884 | 0.906 | 0.825 | 0.863 | 0.011 |
| *Delete_t* | 0.829 | 0.860 | 0.800 | 0.829 | 0.799 | 0.859 | 0.828 | 0.012 |
| *Delete_v* | 0.853 | 0.829 | 0.904 | 0.865 | 0.885 | 0.798 | 0.839 | 0.001 |
| *Deelete_c* | 0.848 | 0.850 | 0.913 | 0.880 | 0.898 | 0.826 | 0.860 | 0.008 |
| *Delete_b* | 0.872 | 0.854 | 0.911 | 0.881 | 0.896 | 0.831 | 0.862 | 0.007 |

**Table 7.** Ablation experiments on Twitter dataset.

| Model | Accuracy | Fake News | | | Real News | | |
|---|---|---|---|---|---|---|---|
| | | Precision | Recall | F-Score | Precision | Recall | F-Score |
| *OUR* | 0.895 | 0.938 | 0.892 | 0.914 | 0.831 | 0.900 | 0.864 |
| *Delete_t* | 0.838 | 0.850 | 0.901 | 0.875 | 0.813 | 0.729 | 0.769 |
| *Delete_v* | 0.867 | 0.849 | 0.959 | 0.901 | 0.911 | 0.711 | 0.799 |
| *Delete_c* | 0.865 | 0.883 | 0.905 | 0.894 | 0.831 | 0.798 | 0.814 |
| *Delete_b* | 0.892 | 0.930 | 0.896 | 0.913 | 0.834 | 0.886 | 0.859 |

### 4.6. Hypothetical Test

The model proposed in this paper is improved on the basis of the AMFB model, and the AMFB model is the best performing model among the above-mentioned benchmark models. Therefore, the *t*-test between the model proposed in this paper and the AMFB model is performed to detect whether the difference between them is significant. First, we hypothesize that the AMFB model is not significantly different from the model proposed in this paper. As can be seen from Table 8, the *p*-value is less than 0.05, so the null hypothesis is rejected, indicating that the AMFB model is significantly different from the model proposed in this paper. As a test, t-values are also looked up based on degrees of freedom and confidence levels to verify that the conclusions are correct. By querying the t-distribution table, when the degree of freedom is 18 and the confidence level is 95%, the t-value is 1.7341, and the t-value in Table 8 is greater than this value, proving that the conclusion is correct.

**Table 8.** The results of *t*-test.

| Compare Models | Twitter | Weibo |
|---|---|---|
| OUR and AMFB | $t = 6.937, p = 5.452 \times 10^{-12}$ | $t = 8.437, p = 9.314 \times 10^{-17}$ |

## 5. Conclusions

The detection model proposed in this paper has the following advantages: (1) It can interact with the data of different modalities in the early stage to exploit its correlation; (2) It can remove the influence of noise during the fusion process; (3) It can reduce the loss of information in the fusion process; (4) The post-fusion module enables the model to handle the asynchrony of the data, allowing different modalities to use their most appropriate analysis method. At the same time, it contains several disadvantages: (1) The remaining information, other than text and images, cannot be used; (2) The extraction of features is difficult. The features of different modalities must have the same format. It is difficult to express the time synchronization between multimodal features, and it is difficult to obtain the cross-correlation between features as the number of features increases; (3) The model structure is complex and the training is difficult; (4) Only the overall fusion is performed on the data of different modalities, and the local fusion is not performed.

This paper proposes a model for false information detection using multimodal information. The model uses a hybrid fusion method to fuse multimodal information, firstly using a deep autoencoder as an early fusion model to obtain multimodal joint features, and the loss of original information in the fusion process is reduced by concatenating text features and image features on the joint features. Finally, the multimodal data are further fused by the late fusion method. The experimental results on Weibo dataset and Twitter dataset show that the detection accuracy of our model is better than the baseline model. At the same time, the model proposed in this paper has limitations: (1) For a post containing multiple pictures, only one of the pictures can be used to detect it, and all the pictures cannot be used at the same time; (2) The complex structure of the model has a large number of parameters and cannot be run on small devices. The following issues should be addressed in future work: (1) Reduce the complexity of the model so that it can be applied to small devices; (2) Use all the information in the post to detect it; (3) Detect false information that has begun to spread on social platforms.

**Author Contributions:** Conceptualization, Y.L.; methodology, Y.L.; software, T.T. and A.H.; validation, Y.L., T.T. and A.H.; formal analysis, Y.L.; investigation, Y.L. and T.T.; resources, T.T.; data curation, T.T.; writing—original draft preparation, Y.L.; writing—review and editing, Y.L. and T.T.; visualization, Y.L.; supervision, T.T.; project administration, T.T. and A.H.; funding acquisition, T.T. and A.H. All authors have read and agreed to the published version of the manuscript.

**Funding:** This work has been supported by the National Natural Science Foundation of China (62166042, U2003207), Natural Science Foundation of Xinjiang, China (2021D01C076), and Strengthening Plan of National Defense Science and Technology Foundation of China (2021-JCJQ-JJ-0059).

**Institutional Review Board Statement:** Not applicable.

**Informed Consent Statement:** Not applicable.

**Data Availability Statement:** Twitter dataset: https://github.com/MKLab-ITI/image-verification-corpus. Weibo dataset: https://drive.google.com/file/d/14VQ7EWPiFeGzxp3XC2DeEHi-BEisDINn/view?usp=sharing (accessed on 27 February 2022).

**Conflicts of Interest:** The authors declare no conflict of interest.

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
