# Peer review of "False Information Detection via Multimodal Feature Fusion and Multi-Classifier Hybrid Prediction"

_algorithms, doi:10.3390/a15040119_

Round 1

Reviewer 1 Report

I suggest that the authors do not use acronyms in the abstract.

The whole pseudocode must be added to the paper. 

Is code somewhere published?

How were the parameters and weights set?

Otherwise, the paper is well-written, but the literature review could be extended. The proposed model and algorithm are well defined, and a reader can easily follow the paper. The results are provided and well discussed. Used references are relevant and up-to-date. After addressing the few mentioned comments, I have no problems with the paper being accepted. 

Reviewer 2 Report

In order to improve the quality of this work, some comments have been given as below.

  1. From literature review of subsection 2.2, using both text and image features is not new. Authors should clarify the unique contribution of this study.
  2. In introduction the research gap between this work and other published studies should be highlighted.
  3. In methodology section, it’s better to describe used methods and equations step by step following Figure 1.
  4. If a false information doesn’t contain image, how can we do by using your method?
  5. In experiments, authors have to use 5-fold or 10-fold cross validation experiment.
  6. The results should be formally analyzed. The differences between models should be testified by using statistically hypotheses.
  7. Additional discussion section should be provided to illustrate the pro and cons of the used method.
  8. The conclusion section should be enriched. Except unique findings, authors should mention the limitations of used method and potential direction of future works.

9.Please cite more the latest literatures.

10.Some format problems are listed as follows.

  • In abstract, authors should provide the full name of BERT.
  • In subsection 2.2 and other paragraphs, authors should use last name only. For examples, authors should revise the citation format “Kumari R [18]” to “Kumari [18]”.
  • The whole section of “experimental results and analysis” should be numbered as 4 not 3.

Round 2

Reviewer 2 Report

1.The related descriptions of proposing your algorithm which have been listed in the end of subsection 2.2 should be mentioned in introduction section to enhance the motivation of this work. Besides, this motivation should be supported by academic evidences, for examples, published works.
2.In methodology section, a psudeo-code algorithm has been added.
3.In subsection 4.2, the definitions of TP, TN, FP, FN should be mentioned in the manuscript.
4.Authors did know the meaning of cross validation experiments. If authors implement 5-fold or 10-fold CV experiments. The results in Tables 6~8 should have mean value and standard deviation for each method.
5.In addition, results in table 10 is also wrong. The differences between methods in tables 6~8. Authors should listed the hypotheses first and provide the p-value for each pair of comparison.
6.The amount of related works about fake news is too large. But, authors didn't provided enough information to let readers know the development of related works. Maybe authors can provide an additional section of fake news detection.

Author Response

Dear reviewer:

Thanks very much for taking the time to review this manuscript. We really appreciate all your generous comments and suggestions! Here are our detailed answers below.

Point 1: .The related descriptions of proposing your algorithm which have been listed in the end of subsection 2.2 should be mentioned in introduction section to enhance the motivation of this work. Besides, this motivation should be supported by academic evidences, for examples, published works.

Response 1:

We feel really sorry for our carelessness.

We mention the relevant description of the algorithm in the introduction section. Academic evidence has also been added to support this motivation.

After modification: Line 58-82, Reference 26, Reference 27

Point 2: In methodology section, a psudeo-code algorithm has been added. 

Response 2:

Thank you very much for your valuable comments.

We have removed table 4 in methodology.

After modification: Table 4

Point 3: In subsection 4.2, the definitions of TP, TN, FP, FN should be mentioned in the manuscript

Response 3:

Thanks for your suggestions.

Our mention the definitions of TP,TN,FP,FN in the manuscript.

After modification: page 9, Line 311-314

Point 4: Authors did know the meaning of cross validation experiments. If authors implement 5-fold or 10-fold CV experiments. The results in Tables 6~8 should have mean value and standard deviation for each method.

Response 4:

Thanks for your help.

Since the dataset used in this paper is a public dataset, the public dataset has been divided into training dataset and test dataset. The experimental results of the benchmark model in Table 5 are all from the original paper. The experimental results of the original paper were obtained by the author through a large number of experiments, which has avoided this problem, so this paper only cross-validates its own model here. Cross-validation was not possible due to the uneven distribution of the Twitter dataset itself. Therefore, we only perform 5-fold cross-validation on the 5 proposed models in Table 6 to ensure the accuracy of the ablation experimental results.

We believe that it is fair to compare the results presented in the original author's paper with the baseline model on the same dataset and with the same partition. The author of the original paper did not provide the source code. If we reproduce the model according to the paper, some details cannot be taken into account, which will also affect the performance of the model. Therefore, we can only use the fairest way of comparison, which is to compare with the results in the original paper.

After modification:Table 6,page 11, Linea 389-393

Point 5: In addition, results in table 10 is also wrong. The differences between methods in tables 6~8. Authors should listed the hypotheses first and provide the p-value for each pair of comparison.

Response 5:

Thank you very much for your suggestion.

Our model is improved on the AMFB model, and compared to the rest of the models, the AMFB model is the best performing model, so we do a t-test to prove whether the difference between our model and the AMFB model is significant.

After modification: page 13,Table 6,Linea 415-426

Point 6: The amount of related works about fake news is too large. But, authors didn't provided enough information to let readers know the development of related works. Maybe authors can provide an additional section of fake news detection.

Response 6:

Thank you very much for your suggestion.

We introduce the development process of false information detection technology in the related work part of Chapter 2. The specific process is as follows:

We divide the models into two categories: single-modal models and multi-modal models according to the chronological order of technology appearance. For the single-modal model, we also introduce it in the chronological order of the technology and in combination with the background of the times. First of all, since the information in the early social media is mostly plain text information, we introduce the method to detect false information from the text aspect. Then, with the development of the times, picture information appeared in social media, so the method of using picture information to detect false information was introduced. With the emergence of new media, information in social media began to contain different forms of information such as text and images. Therefore, the method of detecting false information using multimodal information was introduced. The researchers use the data of both text and image modalities to detect false information at the same time, so the method of using these two kinds of information to detect at the same time is introduced. Subsequent researchers found that the data of the remaining modalities also had an impact on the final results, so we introduced the technique of using more modalities at the same time to detect false information. Finally, the researchers found that the use of different methods to fuse the features of different modalities will also affect the results. Therefore, we introduced the related technologies of using different methods to fuse the features. The model proposed in this paper also belong to the research on this issue.

After modification: page 3, Line 94-96,103-104.

Thank the reviewer for the kind advice again. If you have any questions, please contact us without hesitation.

Yours sincerely

Round 3

Reviewer 2 Report

If you did 5-fold CV experiment, all performance numbers including accuracy, precision, recall, F1 should have average (standard deviation).